# 3D-Printed Collagen Scaffolds Promote Maintenance of Cryopreserved Patients-Derived Melanoma Explants

**DOI:** 10.3390/cells10030589

**Published:** 2021-03-07

**Authors:** Yun-Mi Jeong, ChulHwan Bang, MinJi Park, Sun Shin, Seokhwan Yun, Chul Min Kim, GaHee Jeong, Yeun-Jun Chung, Won-Soo Yun, Ji Hyun Lee, Songwan Jin

**Affiliations:** 1Department of Mechanical Engineering, Korea Polytechnic University, Gyoenggi-do, Siheung-si 15073, Korea; pmj1210@kpu.ac.kr (M.P.); yuntobi@kpu.ac.kr (S.Y.); cmkim@kpu.ac.kr (C.M.K.); wsyun@kpu.ac.kr (W.-S.Y.); songwan@kpu.ac.kr (S.J.); 2Department of Dermatology, Seoul St. Mary’s Hospital, College of Medicine, The Catholic University of Korea, Seoul 296-12, Korea; mrbangga@catholic.ac.kr; 3Precision Medicine Research Center, IRCGP, Department of Biomedicine & Health Sciences, Graduate School, The Catholic University of Korea, College of Medicine, Seoul 296-12, Korea; sunshin@catholic.ac.kr (S.S.); yejun@catholic.ac.kr (Y.-J.C.); 4Department of Mechatronics, Gyeongsang National University, Gyeongsangnam-do, JinJu 52828, Korea; 5Department of Biomedicine & Health Sciences, The Catholic University, Seoul 296-12, Korea; worldgh27@naver.com

**Keywords:** three-dimensional culture system, 3DX printer, 3D-printed collagen scaffolds, patient-derived melanoma explants, cryopreserved biospecimens

## Abstract

The development of an in vitro three-dimensional (3D) culture system with cryopreserved biospecimens could accelerate experimental research screening anticancer drugs, potentially reducing costs and time bench-to-beside. However, minimal research has explored the application of 3D bioprinting-based in vitro cancer models to cryopreserved biospecimens derived from patients with advanced melanoma. We investigated whether 3D-printed collagen scaffolds enable the propagation and maintenance of patient-derived melanoma explants (PDMEs). 3D-printed collagen scaffolds were fabricated with a 3DX bioprinter. After thawing, fragments from cryopreserved PDMEs (approximately 1–2 mm) were seeded onto the 3D-printed collagen scaffolds, and incubated for 7 to 21 days. The survival rate was determined with MTT and live and dead assays. Western blot analysis and immunohistochemistry staining was used to express the function of cryopreserved PDMEs. The results show that 3D-printed collagen scaffolds could improve the maintenance and survival rate of cryopreserved PDME more than 2D culture. MITF, Mel A, and S100 are well-known melanoma biomarkers. In agreement with these observations, 3D-printed collagen scaffolds retained the expression of melanoma biomarkers in cryopreserved PDME for 21 days. Our findings provide insight into the application of 3D-printed collagen scaffolds for closely mimicking the 3D architecture of melanoma and its microenvironment using cryopreserved biospecimens.

## 1. Introduction

Melanoma is a highly aggressive skin cancer. It can grow deep into the skin layer and spread to other parts of the body very rapidly if it is not treated at an early stage [1,2]. Melanoma therapy choices depend on the stage of the cancer, the location of the tumor, and the patient’s health. Melanoma that is discovered at an early stage is detected in the upper layers of the skin and without any indication of further spreading [1,2]. Early-stage melanoma can be removed easily via surgery. Stage III and stage IV melanoma have metastasized beyond the original tumor to the lymph nodes and/or distant organs, making it much more difficult to remedy with current treatments [3]. Although melanoma accounts for less than 1% of all skin cancers, it is responsible for 80% of deaths from all skin cancers [3]. Melanoma is a highly heterogeneous and mutated subpopulation of tumor cells [4]. Treated patients with advanced melanoma experience long-term side effects and quickly emerging resistance to targeted therapies [4,5].

Advanced melanoma is extremely challenging to prevent, detect, and monitor, as well as being resistant to therapy [5]. To explore how melanoma evades targeted therapies, patient-derived melanoma xenograft (PDX) was developed through transplanting surgical patient-derived tumor specimens or well-validated cancer cell lines into immunocompromised mice [6,7,8,9]. In general, traditional two-dimensional (2D) patient-derived primary melanoma cells and melanoma cell lines do not necessarily reflect the molecular and biological properties of the original melanoma in patients [1]. The PDX model is known as a patient “avatar” [6,7,8,9]. PDX models have been demonstrated to more closely resemble the original tumor due to the level of heterogeneity and the properties of the microenvironment, retaining the cellular complexity, cytogenetics, and stromal architecture maintained in the in vivo system [6,7,8,9]. However, a limitation of PDX models is the possibility of original human stromal cells in tumors dissected from patients being gradually replaced by murine stromal cells as the xenograft grows.

Three-dimensional (3D) in vitro cancer models and 3D bioprinting have emerged as highly desirable tools for analyzing the roles of various biochemical and biophysical cues in tumor initiation and progression [10]. 3D bioprinting, an additive manufacturing technique, refers to the creation of live tissue models using a 3D printer [11,12]. In principle, 3D bioprinting involves choosing the 3D printer and structural designs, selecting a printable bioink, defining the cell types and densities, performing post-fabrication tissue support and finally, integrating all these steps to obtain a viable and functional tissue [11,12]. However, bridging between 3D bioprinting-based 3D in vitro cancer models and cryopreserved biospecimens derived from patients with advanced melanoma remains challenging. The aim of this study was to explore whether 3D-printed collagen scaffolds can support a 3D culture system for propagation and maintenance of cryopreserved patient-derived melanoma explants (PDMEs). The present study was performed to characterize the expansion of cryopreserved PDME by 3D-printed collagen scaffolds in vitro within short and/or long-term conditions.

## 2. Materials and Methods

### 2.1. Cell Line and Reagents

Cryopreservation freezing solution was obtained from ZENOAQ RESOURCE CO., LTD (Fukushima, Japan). DAPI, Live and dead viability/cytotoxicity kit, and MTT assay kit were purchased from Thermo Fisher Scientific (Rockford, IL, USA). MS Collagen (Type 1 atelo-collagen from porcine skin) was obtained from MSBio, Inc. (Gyeonggi, Korea). Antibodies that recognize MITF (ab20663), Mel A (ab731), H2B (ab52599), S100 (ab4066), and tyrosinase (ab738) were purchased from Abcam (Cambridge, UK). Antibodies specific for actin (sc47778) were obtained from Santa Cruz Biotechnology, Inc. (Santa Cruz, CA, USA). A375 melanoma cell lines were obtained from the American Type Culture Collection (ATCC, Manassas, VA, USA). A375 cells were cultured in DMEM-high glucose with 10% FBS and 1% PS (penicillin-streptomycin).

### 2.2. PDME, PDX Model and Cryopreservation

Melanoma tissues were collected from two melanoma patients at Seoul St. Mary’s Hospital (Seoul, Korea) with the approval of the institutional review board (IRB:KC15TISI0966) (Appendix A). Fresh melanoma samples were directly implanted on the back of immunodeficient NOD/SCID/IL-2R γ null (NSG) mice under the ethics protocol approved by the Institutional Animal Care and Use Committee of the Catholic University of Korea. According to the Institutional Animal Care and Use Committee of the Catholic University of Korea’s standardized pain protocol, all mice were continually monitored for signs of distress. To harvest melanoma PDX (MPDX) and normal skin (NS), mice were euthanized 54 days after implantation of melanoma tissue (Appendix A). Engrafted MPDX in Hank’s Balanced Salt solution (HBSS) buffer with penicillin and streptomycin (PS) were sliced into 5 × 2 mm pieces with a sterile scalpel blade. The sliced MPDX (1 per vial) were placed in plastic screw top cryopreservation vials pre-embedded with 1.5 mL of cryopreservation freezing solution. These vials were placed into a freezing container of 100% isopropyl alcohol, and then were subsequently kept in a −80 °C deep freezer for at least 24–48 h before transfer to a liquid nitrogen tank.

### 2.3. Fabrication of 3D-Printed Collagen Scaffold-on-Frame Construction and Loaded with Cryopreserved PDME

To generate uniform and stable 3D-printed collagen scaffolds, we used the method of extrusion printing with 3DX Printer (T&R Biofab Co. Ltd, Siheung, Korea) [13]. Briefly, 3% collagen solution was loaded into a syringe and printed as filaments onto a stage via a mechanical dispensing system. Precise deposition of collagen was controlled by a dispensing stage along the x, y, and z axes. Our method of collagen printing accommodated a broad range of ink viscosities (30–60 × 10^7^ mPa·s), the print resolution (500 μm), and the nozzle diameter. For gelation, collagen scaffolds-on-frame construction incubated at 37 °C in a 5% CO_2_ incubator for 10 min. The weight of frame constructions before and after 3D-printed condition were measured using an electronic balance (model GR-200 and CUX220H). Simultaneously, deep frozen cryovials that were maintained in a liquid nitrogen tank were thawed through a water bath at 37 °C for 1 min. The cryopreserved PDMEs were washed with HBSS buffer (at least twice) to remove all cryoprotectant solvents. After thawing, cryopreserved PDMEs were cut into approximately 1–2 mm fragments to load with 3D-printed collagen scaffolds. 3D-printed collagen scaffold-on-frame constructions were loaded with one fragment of cryopreserved PDME. For 2D culture, cryopreserved PDMEs were cut into approximately 1–2 mm fragments. A total of 30 samples (24 for survival and Western blot analysis; 6 for imaging) of these fragments and 3D-printed collagen scaffold-on-frame constructions were cultured in a growth medium of α Minimum Essential Medium (MEM) with 10% Fetal Calf Serum (FCS), 2 mM L-glutamine, and 1% PS. They were incubated at 37 °C in a 5% CO_2_ incubator.

### 2.4. The Viability and Proliferation Assay

2D-culture of cryopreserved PDME and 3D-printed collagen scaffolds embedded with cryopreserved PDME were cultured in a growth medium at the short-term (7 days) or long-term culture condition (21 days). We changed the culture medium every 2 days. At the indicated days, the viability and proliferation of the cryopreserved PDME were assessed using an MTT assay kit. The samples were incubated for 2 h in a culture medium with MTT solution. The incubated culture medium was removed, and then was extracted with DMSO for 1 h at RT. We used a negative control which had only DMSO without cryopreserved PDME. Absorbance was determined at 590 nm using an ELISA reader (Emax; Molecular Devices, Sunnyvale, CA, USA).

### 2.5. LIVE/DEAD Staining

To visualize the viability of the cryopreserved PDME, a double staining kit of live and dead cells was used, combining calcein- acetoxymethyl (AM) (used for fluorescent staining of living cells) and propidium iodide (PI, used for fluorescent staining of dead cells) [13]. Cryopreserved PDMEs were placed in the staining solution (calcein-AM with PI), and then incubated at 37 °C for 30 min with protection from light in a 5% CO_2_ incubator. Images were acquired using an Olympus FV1200 confocal microscope with 405, 473, 559, and 635 nm laser lines.

### 2.6. Flow Cytometric Analysis

Fluorescence Activated Cell (FACS) analysis further confirmed the expression of targeted proteins (tyrosinase and MITF as melanoma marker; H2B as a marker of living cells and potential biomarker of epigenetic target for melanoma) in cryopreserved PDME. Thawed PDMEs were enzymatically digested with fresh digest media (200 U/mL Collagenase IV, 5 mM CaCl_2_, 50 U/mL DNase in HBSS ^-/-^ followed by [6]. After fixing in 4% paraformaldehy(PFA) at 4 °C, cells were washed with PBS, permealized with 5% Bovine Serum Albumin (BSA) and 0.1% Triton X-100, and incubated with primary antibodies. FACS analysis was performed using a CytoFLEX flow cytometer (Beckman coulter Life Sciences).

### 2.7. Immunohistochemistry (IHC) and Immunofluorescence Staining (IFS)

The samples were fixed with 4% formaldehyde, embedded with paraffin, and sectioned into 4-μm-thick sections (IHC) or 20 μm-thick sections (IFS). The slides were stained with hematoxylin and eosin (H&E) or IFS according to standard methods [14]. IFS-labelled slides were incubated with primary antibodies (1:100). After nuclear DAPI staining, immunostained cells were imaged using Olympus FV1200 confocal microscope with 405, 473, 559, and 635 nm laser lines. For IHC staining, heated target retrieval solution was used, and the slides were incubated by peroxidase-blocking solution for 15 min at room temperature (RT). Primary antibodies (1:100 dilution, Mel A, an antigen on melanocytic tumors) were incubated overnight at 4 °C in humid chamber. Horseradish peroxidase (HRP)-conjugated secondary antibody was detected using the Real Envision system kit (Dako, Carpinteria, CA, USA) for 30 min at RT. The immunoreaction was developed for one min, and hematoxylin counterstaining was used. Brightfield photographs were captured using Leica light microscope DMI 5000B (Leoca, Wetzlar, Germany).

### 2.8. Western Blot Analysis

The frozen samples were disrupted using the homogenizer, after which an ice-cold PRP-PREP protein extraction solution with a protease inhibitor cocktail (iNtRON Biotechnology, Inc., Seoul, Korea) was added, and the samples were homogenized by a microtube homogenizer system (PRO, Scientific) [15]. Protein concentration was assessed using a BCA-kit (Thermo Scientific, Rockford, IL, USA). An equal amount of protein (80 μg) from each sample was loaded onto 10% to 12% SDS gel, and transferred to a PVDF membrane. The membranes were blocked for 2 h at RT with 5% BSA in PBS containing 0.1% Tween-20, and incubated with primary antibodies (1:1000 and 1:500) overnight at 4 °C. After washing three times, the membranes were incubated with a HRP-conjugated secondary antibody (1:5000) at RT for 2 h and visualized with a chemiluminescence substrate.

### 2.9. Statistical Analysis

Student’s *t*-tests (for comparisons of two groups) were used for the statistical analyses. SPSS software ver. 17.0 (SPSS, Chicago, IL, USA) and GraphPad Prism software (GraphPad Software, SanDiego, CA, USA) were used to perform the statistical analysis. Data are expressed as means ± standard error of the mean (SD). A value of *p* < 0.05 was considered significant. * *p* < 0.05–0.01, ** *p* < 0.01–0.001, and *** *p* < 0.001 vs. corresponding controls. All error bars represent the standard deviation of three or more biological replicates.

## 3. Results

### 3.1. The Viability and Characterization of Cryopreserved PDME after Thawing

To investigate the screening of candidate drugs for potential antimelanoma action, an MPDX model can be used for the propagation and expansion of patient tumors by implanting tumor fragments or single cells into immunodeficient NSG^TM^ mice [6,7,8]. However, minimal research utilizes MPDX models due to their considerable expense and prolonged time frame (often over 2 months) required for initial set-up. To provide a more practical alternative for researchers conducting preclinical drug evaluations, we sought to integrate the use of 3D bioprinting and PDX modeling. Baseline patient melanoma generally undergoes reanimation engraftment after cryopreservation [16,17]. The efficacy of reanimating cryopreserved primary melanoma patient tumors and their PDX has typically been low or not investigated [16,17]. To confirm this low efficacy, we established MPDX models (Figure 1A). We put 10–20 mm fragments of sliced MPDX (1 piece per vial) in plastic screw top cryopreservation vials pre-embedded with 1.5 mL of cryopreservation freezing solution. These vials were kept in a liquid nitrogen tank (Figure 1A). Three randomly selected vials of cryopreserved PDME after thawed were assessed by MTT assay, live and dead assay, and FACS analysis to characterize and determine the viability of the cryopreserved PDME. MTT assay demonstrated that there was an approximately 4-fold increase in cryopreserved PDME compared to DMSO (Figure 1B). Live and dead assay also revealed that the number of green-labeled cells was higher than red-labeled cells in the cryopreserved PDME (Figure 1C). FACS analysis further confirmed that expression of H2B was elevated in the cryopreserved PDME but not in the control (Figure 1D). The expression of tyrosinase and MITF in the cryopreserved PDME increased (Figure 1D).

### 3.2. Differences in 2D vs. 3D-Printed Collagen-Scaffolds-on-Frame Construction for Short-Term Maintenance of Cryopreserved PDME

Recently, various types of 3D bioprinting technologies, processes, and their printable collagen inks can be designed and developed in biomimetic tissue constructs with polymeric frameworks and multiple biochemical and biophysical cues [18]. If combining cryopreserved PDME and 3D printing methods in an in vitro tri-culture model can improve the survival of cryopreserved PDME, many researchers could potentially use 3D bioprinting to generate tumor models. To test this, using a 3DX bioprinter, we created a collagen scaffolds-on-frame-construction via layer-by-layer deposition of collagen I (Figure 2A). Fragments of cryopreserved PDME were embedded in the collagen scaffolds-on-frame-construction for an in vitro 3D culture model of melanoma. The 3D-printed collagen scaffolds with cryopreserved PDME were incubated for 7 days. Representative confocal live and dead assay images showed enrichment of a green-fluorescent calcein labeling of cryopreserved PDME, indicating a high survival rate of cells in 3D-printed collagen scaffolds (Figure 2B). To confirm this observation, MTT assay revealed that the survival rate of cryopreserved PDME at 7 days significantly increased by 4-fold in 3D-printed collagen scaffolds compared to the 2D culture method (Figure 2B,C). Collagen scaffold-based microenvironments are known to have an abundance of physiological conditions that support the survival and proliferation of melanoma in a paracrine fashion [19]. These results provide evidence that 3D-printed collagen scaffolds might accelerate the expansion of cryopreserved PDME, boosted by an interaction with a cell-collagen scaffold microenvironment.

### 3.3. The Uniformity of Morpholgy and the Shape stability of 3D-Printed Collagen-Scaffolds-on-Frame Construction for Subsequent Long-Term Maintenance of Cryopreserved PDME

Next, we determined whether 3D-printed collagen scaffolds are associated with the physiological condition of cryopreserved PDME to allow melanoma research in a microenvironment that more closely resembles in vivo and is less time-consuming than animal studies. According to the principle of hydrogel-based 3D bioprinting, collagen-based ink can be arranged in a tunable shape when incubated at 37 °C, allowing collagen to form stable gelation after extrusion printing [13,18,19]. Optical images of the 3D-printed collagen-scaffolds-on-frame constructions and related graphs show a constant weight and a shape fidelity post printing (Figure 3A). Diameter and thickness of 3D-printed collagen-scaffolds-on-frame constructions maintain a structural stability until 21 days (Figure 3B,C). During the cultivation of 21 days, the size of the cryopreserved PDME expanded in the 3D-printed collagen scaffolds. These data suggest that this 3D model can be effectively used to maintain and expand cryopreserved PDME.

### 3.4. 3D-Printed Collagen Scaffolds Help to Maintain the Functional Health of Cryopreserved PDME

To better understand the effect of 3D-printed collagen on the culture of cryopreserved PDME, we evaluated the functional health of cryopreserved PDME during its cultivation at the indicated time points using MTT assay, Western blot analysis, and IHC staining. 3D-printed collagen scaffolds loaded with cryopreserved PDME after thawing showed higher expression of MITF, Mel A, and S100 than found in NS and the A375 cells (Figure 4A). As shown in Figure 4B, optical images of 3D-printed collagen scaffolds reveal the expansion of cryopreserved PDME in a time-dependent manner. Consistent with this observation, MTT assay demonstrated that the percentage of cryopreserved PDME survival in 3D-printed collagen scaffolds dramatically increased approximately 3-fold in a time-dependent manner compared to the 2D culture method (Figure 4B). Western blot analysis further confirmed that MITF proteins were stably expressed in 3D-printed collagen scaffolds (Figure 4C). Furthermore, the 3D-printed collagen scaffolds led to a significant increase in S100 and Mel A expression as well as in the size of the cryopreserved PDME (Figure 4D and Appendix A). Taken together, these findings indicate that our protocol can generate a healthy 3D culture of melanoma that more accurately mimics the PDME microenvironment in vitro.

## 4. Discussion

In the present study, our findings highlight the significant role that 3D-printed collagen scaffolds can play in supporting 3D in vitro cancer models with cryopreserved PDME. A comparative analysis of 2D cultures and 3D-printed collagen scaffold-based cultures revealed that the latter is more effective in the maintenance of cryopreserved PDME for both in vitro and ex vivo cultures. We have identified stably expressed four-marker signatures of melanoma in 3D-printed collagen scaffolds with cryopreserved PDME models over long-term conditions. Furthermore, cryopreserved PDME grew around the frame construction. We set out to examine whether 3D-printed collagen scaffolds can contribute to maintaining cryopreserved PDME in an environment straddling in vitro and ex vivo conditions for rewiring melanoma-tumor microenvironment interactions to enable customized melanoma management. It is already known that melanoma and the tumor microenvironment are a complex. Most basic research is still performed in vitro by growing isolated melanoma cells and/or immortalized cell lines in 2D cultures on plastic culture dishes under conditions without the appropriate tumor microenvironment [20,21,22].

The tumor microenvironment, with its cellular and molecular heterogeneity, has a critical impact on tumor progression, metastasis formation, and invasion into other tissues in the human body [4,5]. To better simulate this microenvironment, our method could serve as a halfway point between in vitro and ex vivo alternatives for the effective maintenance of cryopreserved PDME. In adherent 2D culture, cells grow as a monolayer in a culture dish [20,21,22]. Monolayer culture is not sufficient to manifest all of the many complex properties of the melanoma microenvironment in vivo [20,21,22]. In vitro 3D organotypic melanoma spheroids are instead better able to portray the in vivo architecture of malignant melanoma [23,24]. However, nutrient and/or oxygen consumption and metabolite gradients in the outer and inner halves of the viable rim in in vitro 3D organotypic melanoma spheroids become insufficient to maintain the growth over a long culture. In addition, these models require especially labor-intensive and time-consuming manual processing, as well as often involving the expensive consumption of specialized reagents. In contrast, our model capitalizes on automation to substantially reduce costs and time, but the individual characteristics of melanoma patients can be well represented.

In general, reciprocal interaction between fibrillar collagens and melanoma development tumor cells are linked to various types of extracellular matrix, with collagen [25,26]. A previous paper has demonstrated that collagen abundance governs melanoma phenotypes through lineage-specific microenvironment sensing [25]. Using data from TCGA melanoma dataset via cBioportal, which contains expression data from 458 patients of various stages of melanoma, the researchers found that high collagen expression is associated with poorer patient survival, especially when there is also expression of MITF target genes including TRPM1, TYP, TYRP1 (differentiation markers) [25]. Collagen has also been found to influence tumor microenvironment and cancer cell activity [26]. For example, type I collagen expression is correlated with angiogenesis and the development of deeply invasive cutaneous melanoma [26]. Based on these observations, we explored whether 3D-printed type I collagen scaffolds is closely connected to the maintenance and survival rate of cryopreserved PDME. Nonetheless, our study has some limitations of note. One limitation is that it only indirectly tracked the effects of 3D-printed collagen scaffolds on 3D in vitro cancer models with cryopreserved PDME. There remain unresolved questions about the biological significance of 3D-printed collagen scaffolds embedded in the melanoma behavior after transplantation. Moreover, it remains controversial whether all cells derived from cryopreserved PDME maintain their original condition and accurately depict the microenvironment of melanoma in patients [1,2,5,27]. Additional studies based on collagen-mediated melanoma behavior in mice and in patients should be able to provide greater clarity [27,28]. Therefore, the effect of 3D-printed collagen scaffolds on 3D in vitro cancer models with cryopreserved PDME should be further investigated to confirm the precise relationship between 3D-printed collagen scaffolds and the melanoma microenvironment.

## 5. Conclusions

The 3D-printed collagen scaffolds embedded with cryopreserved PDMEs exhibit a higher survival rate and retainment of melanoma biomarkers compared to 2D culture. Additional experiments treating 3D-printed collagen PDME with inhibitors could be performed to evaluate in more detail apoptosis and proliferation of melanoma cells. Our findings offer a useful tool, 3D-printed collagen scaffolds, for the maintenance and expansion of cryopreserved PDME in short and/or long term cultures. It could facilitate rapid screening of the efficacy of new drugs, as well as reduce the expense and need for animal research.

## Figures and Tables

**Figure 1 cells-10-00589-f001:**
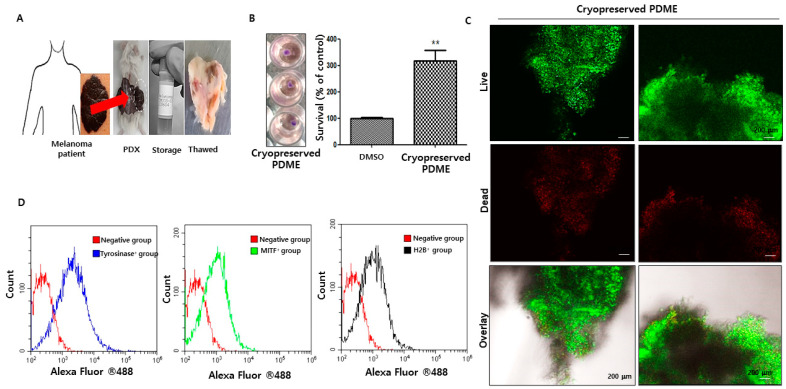
The viability and characterization of cryopreserved patient-derived melanoma explant (PDME( after thawed of frozen samples. (**A**). Images of melanoma patient (**left**), a NSG-based patient-derived melanoma xenograft (MPDX) model (**middle**), liquid nitrogen storage (**middle**), and thawed PDME in HBSS buffer (**right**). After liquid nitrogen storage, the frozen PDME samples were quickly thawed in a 37 °C water bath. Scale bars, 1 cm. (**B**). Digital images of MTT-treated cryopreserved PDME and graphs of the survival rate of cryopreserved PDME after the thawing of frozen samples. The survival rate was determined by the MTT assays. ** *p* < 0.01 versus DMSO without cryopreserved PDME as a negative control using Student’s *t*-test. (**C**). Confocal images independently showing live (green) and dead (red) cryopreserved PDME. Thawed PDME were labeled using a live and dead kit. A confocal microscope was used to detect the live (green) and dead (red) cells. Scale bars, 200 μm. (**D**). Thawed PDMEs were enzymatically digested with collagenase Type IV as described in Materials and Methods. Isolated single cells were labeled with targeted antibodies, and then subjected to CyoFLEX FACS analysis. Fluorescence of tyrosinase^+^ (blue), MITF^+^ (green), and H2B^+^(black) cells was evaluated on a histogram of CytoFLEX FACS acquisition and analysis. The negative groups (red) indicate as the assessment of nonspecific binding of Alexa Fluor^®^488 antibodies to single cells derived from cryopreserved PDME.

**Figure 2 cells-10-00589-f002:**
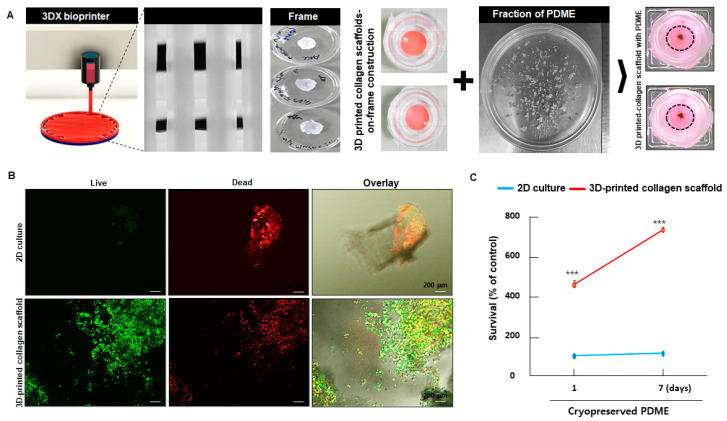
The effects of 3D-printed collagen scaffolds on 3D culture model of cryopreserved PDME. (**A**) Manufacturing process of 3D-printed collagen scaffolds loaded with cryopreserved PDME. Two layers of collagen-on-frame construction were printed by a 3DX bioprinter. (**B**) Confocal images for visualizing live (green) and dead (red) labeled cells in 2D-cultured-and 3D-printed collagen scaffolds loaded with cryopreserved PDME. Scale bars, 200 μm. (**C**) Graph to calculate the proliferation of 2D-cultured and 3D-printed collagen scaffolds loaded with cryopreserved PDME after the indicated time points. All data were obtained from three independent experiments, and values represent the mean ± SD. For all groups, *** *p* < 0.001 versus 2D-cultured at the indicated time points using Student’s *t*-test.

**Figure 3 cells-10-00589-f003:**
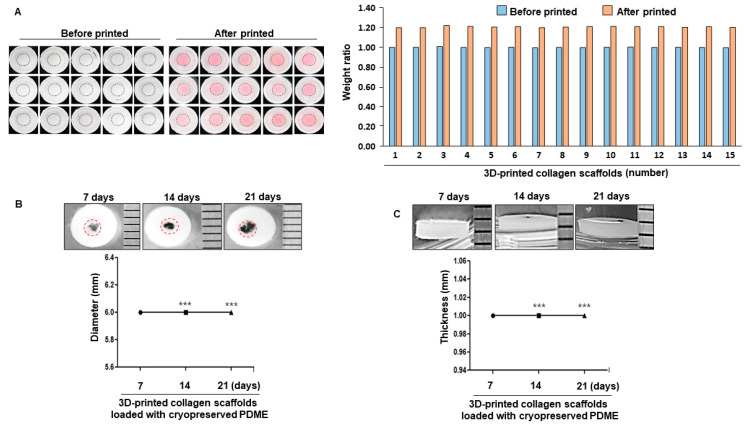
Long-term stability of 3D-printed collagen scaffolds loaded with cryopreserved PDME in a time-dependent manner. (**A**) Representative images showing 3D-printed collagen scaffold-on-frame constructions. The graph indicates the uniform weight of collagen scaffold-on-frame constructions before the 3D-printed group (blue box) and after the 3D-printed group (red box). (**B**) Diameter images of 3D-printed collagen scaffold loaded with cryopreserved PDME at the indicated time points. The graphs indicate the diameters of the respective groups. Red circles show cryopreserved PDME. (**C**) Thickness images of 3D-printed collagen scaffolds loaded with cryopreserved PDME at the indicated time points. The graph depicts thickness measurements. The diameter and thickness were measured with a ruler. Scale bars, 1 mm. All data were obtained from three independent experiments. *** *p* < 0.001 versus 3D-printed collagen scaffolds-loaded with cryopreserved PDME at 7 days using Student’s *t*-test

**Figure 4 cells-10-00589-f004:**
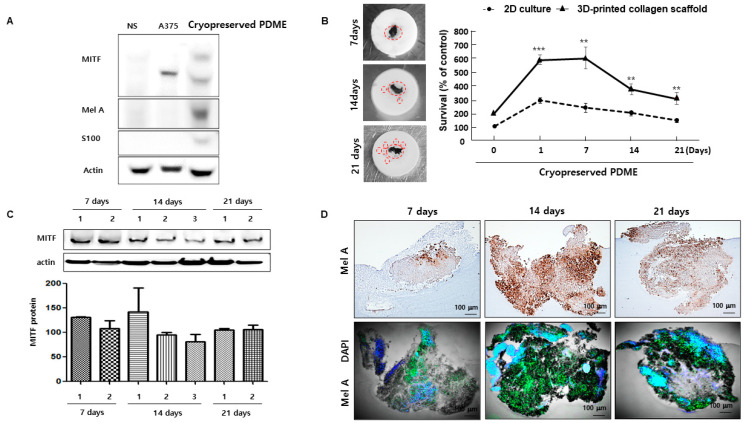
The effects of 3D-printed collagen scaffolds on 3D culture model of cryopreserved PDME in the maintenance of long-term culture. (**A**) Western blot analysis to detect the expression of MITF, Mel A, and S100 indicating the expression of NK, A375, and 3D-printed collagen scaffolds loaded with cryopreserved PDME after thawing. (**B**) Representative images of 3D-printed collagen scaffolds loaded with cryopreserved PDME at 7, 14, and 21 days. The graphs indicate the proliferation of the 2D-cultured and 3D-printed collagen scaffolds at the indicated time points. Red circles show cryopreserved PDME and its satellite. All data were obtained from three independent experiments, and values represent the mean ± SD. For all groups ** *p* < 0.01, *** *p* < 0.001 versus 2D-cultured at the indicated time points using Student’s *t*-test. (**C**) Western blot analysis for expressing MITF protein at the indicated time points. The graph indicates densitometry and statistical analysis of MITF protein expression gathered from twice experiments of western blot assay. (**D**) Immunohistochemisty (IHC) and immunofluorescence staining (IFS) analysis for maintaining the function of 3D-printed collagen scaffold laded with cryopreserved PDME at the indicated time points. Mel A(green), DAPI(blue), Overlay of Mel A and DAPI(bright blue). Scale bars, 100 μm.

## Data Availability

All results generated or analyzed during present study are included in this published article. Data and materials will be made available upon request via email to first author (phdjeongym12@kpu.ac.kr).

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
