# Peer review of "3D-Printed Collagen Scaffolds Promote Maintenance of Cryopreserved Patients-Derived Melanoma Explants"

_cells, 2021, doi:10.3390/cells10030589_

Round 1

Reviewer 1 Report

General Comments

The authors present an interesting paper that explores whether 3D printed collagen scaffolds can serve as a rapid and cost-effective tool for the expansion of PDX derived explants for downstream assays. In general, the paper is lacking information about disease biology and whether this system can assist in the study of skin melanoma in all its forms. For example, the reader would be interested to know the pathology of the two skin samples analysed; what stage of disease were they from, particularly as the authors themselves mention that this is a heterogeneous cancer. Why was collagen I selected – is this relevant to the ECM in the skin lesions? More detail is needed in the Materials and Methods and Results sections so it is clear how many samples were analysed in each part of the study.

Specific Comments:

  • Line 99, please describe the criteria by which mice were euthanized at day 55-65. Was this based on size of a palpable tumour in the mice?
  • Line 112 – a broad range of ink viscosities are detailed – how is this relevant, perhaps to forces/tensions found in the melanoma samples? What are the mechanical properties of the collagen scaffold and how does this compare to skin melanoma?
  • How can the authors be sure that the MTT, calcein-AM and PI, were given enough time to penetrate the denser 3D models? Please provide more details on these methods.
  • Line 155 – protein loading for WB is high at 80µg?
  • Please explain why the antibodies were selected – what are they expected to show e.g. H2B? How does this compare with the patient tumour and PDX?
  • Page 5, Figure 1D – the negative control % in the bold box is detailed as 49.52%; this is incorrect?
  • Line 185 – cryopreserved PDME compared to DMSO? Please explain exactly what this control is as it is unclear in the text.
  • It is unclear from the text how many pieces of tissue from the two melanoma patients were analysed for each assay and thus for the statistical analyses undertaken? What is shown in each graph should be made clearer to the reader.
  • Lines 199 – 202, CyoFLEX FACS is not described in the Methods.
  • Figure 2C is not a bar graph – all parts of the Figure legend need more information so that it is clear what was done and what is being compared.
  • Please explain Figure 2C - Why is proliferation in the tissue/explant so much higher in the 3D setting at day 0 as compared with that in the 2D setting? Only the change in proliferation over time in each setting should be compared.
  • Line 232 – Please clarify this statement “mean ± SD of three independent experiments in triplicate assays”?
  • Figure 3 is irrelevant to the take home messages of the paper and should be removed.
  • Figure 3D shows no change in diameter over time for a single explant, however in Figure 4D, an explant that has clearly grown is shown. Please explain what the circles are, which are used in Figures 3 and 4. Again how many explants were examined? The data should be presented for all cultures.
  • Figure 4A – MITF western blot shows different size bands in A375 and C-PDME. Please explain? The Western blot should have size markers
  • Line 287 - points not pointes
  • NK detailed in Figure 4A – do the authors mean NS?
  • The Discussion is lacking clear information about what this study has achieved for the field of cutaneous melanoma and translational research. Please describe this and discuss how this technology is being used for other cancers.

Author Response

We really appreciate your review and the thoughtful efforts that the reviewers have expended to aid us in improving our manuscript. According to the reviewers’ comments, we have corrected misspellings and errors in the revised manuscript and indicated these and other revisions accordingly in the tracked manuscript with blue type. The Reviewer’s specific comments are cited in italics, and our detailed responses follow.

Reviewer 2 Report

The authors of the paper “3D-Printed Collagen Scaffolds Promote Maintenance of Cryo-preserved Patients-Derived Melanoma Explants” provide an interesting approach based on 3D-printed collagen scaffolds, to improve the maintenance and survival rate of cryopreserved patient-derived melanoma explants (PDMEs) respect to 2D culture. To this aim, they seeded onto the 3D-printed collagen scaffolds fragments derived from cryopreserved PDMEs (approximately 1-2 mm) and incubated for 7 to 21 days, showing a better survival rate and retainment of melanoma biomarkers respect to 2D culture. Thus, they propose this model as a useful tool for the maintenance and expansion of cryopreserved PDME in short and / or long-term cultures.

Although potentially interesting, this work is not adequate for publication in the present form, but need some revisions.

First, the study is based on the analysis of two melanoma tissues collected from two patients (line 92), without any information about tumor localization and stage. The results obtained would be strengthened by analyzing a larger number of patient-derived melanoma explants that are as homogeneous as possible.

Furthermore, some aspects of experimental data have to be improved and/or better explained.

Figure 1:

  1. B) Authors should explain what they refer with DMSO sample (lines 184/5).
  2. C) By using live and dead assay the authors demonstrate the number of green-labeled cells was higher than red-labeled cells in the cryopreserved PDME. Confocal images are not good, you probably need to show higher magnification alongside. Furthermore, co-localization staining with a specific nucleus dye is preferable for determining the number of positive cells versus total ones for both live and dead staining.

D-G) The figure legend refers to the Materials and Methods for the description of PDME digestion and FACS analysis, but no mention is reported there (lines 199/200). It is not clear what are the “negative groups” indicated by authors (line 202/3).

Figure 2:

  1. B) As already suggested for Figure 1C, the staining colocalization for either live or death cells with a nuclear dye and the evaluation of the positive cell number versus the total number of cells is necessary to demonstrate viability and death differences between 3D-printed collagen scaffold and 2D culture. As regard the latter, it is not clear the procedure to obtain and culture them from cryopreserved PDME (lines 120/1).

Figure 3:

The expansion of cryopreserved PDME size cultured onto the 3D-printed collagen scaffolds (as stated in lines 244/5) needs to be measured over the time and graphed.

Figure 4:

  1. Western blotting results should be supported by densitometry and statistical analysis. There is an incongruence between text and figure, in that in the first is reported NS (line 266), whereas in the latter it is reported NK (line 282), in both cases without explaining what does it mean. The same is true for C-PDME meaning.
  2. The survival rate of 3D-printed collagen scaffold slows down starting from day 7 of culture. If this aspect is due to an increase in the apoptotic index should be investigated.
  3. Authors should indicate samples reported in the histogram of Mitf expression.

Author Response

(The authors gave the same response as above.)

Reviewer 3 Report

Your article “3D-Printed Collagen Scaffolds Promote Maintenance of Cryo-preserved Patients-Derived Melanoma Explants” demonstrates that cryopreserved patient-derived melanoma explants can be supported in your developed 3-D printed collagen scaffolds.

The cryopreservation freezing solution you used was from ZENOAQ. What does this freezing solution use in place of DMSO?

Author Response

We really appreciate your review and the thoughtful efforts that the reviewers have expended to aid us in improving our manuscript.  The Reviewer’s specific comments are cited in italics, and our detailed responses follow.

Reviewer 4 Report

The paper by Jeong et al, is an interesting study showing how 3D-printed collagen scaffolds can be used to prolong the survival of cryopreserved PDME. Methodologically the paper is complete and well describes this useful technology. I suggest to the authors to change images of western blot analysis (Fig. 4A and C), actin is not homogeneous between samples. Moreover add confocal staining for melanoma markers, not only IHC. Laslty, it could be usefull to perform some experiments treating 3D-collagen PDME with BRAF inhibitors and evaluate apoptosis and proliferation of melanoma cells, as a rational to exploit this technology to test novel drugs.

Author Response

(The authors gave the same response as above.)

Round 2

Reviewer 1 Report

The authors have added in useful information about the choice of collagen I for their 3D gels, however, this would be better placed in the Introduction rather than Discussion.

I still feel that it is not clear to the reader how many samples were analysed in each experiment and thus what the data presented actually represent in terms of numbers. The authors now mention 30 samples in the Materials and Methods but again in the Results (for example in the figure legends) it is unclear how many technical or biological replicates were undertaken i.e. n=? in the data shown.

The authors have included the details of FACS analysis under the section relating to LIVE/DEAD staining in the Materials and Methods, however the FACS analysis examines melanoma markers and not viability. The FACS methodology should thus be a separate method.

The authors provide a nice rationale in their comments for selecting a H2B antibody. It would be useful in the Materials and Methods section for the authors to include a small sentence in the FACS analysis where they mention the antibodies to state why they were selected/what they will detect.

Figure 1D, E, F and G now all have black boxes in the antibody +ve parts of the figure – please correct this. Also, the data for 1D must be incorrect as the IgG antibody -ve portion of the figure is >90% and thus the box showing IgG antibody positivity must be <10%.

Since Figure 3 is representing the long-term stability of 3D printed collagen scaffolds please add in the time points at which the two measurements were made for clarity.

I am unclear why NS would be negative for Melan A expression in the WB of Figure 4A. The data in Figure 4A seem inconclusive in what is being shown?

Author Response

We appreciate your review. According to the reviewers’ comments, we have added some words in the revised manuscript and indicated these and other revisions accordingly in the tracked manuscript with blue type. The Reviewer’s specific comments are cited in italics, and our detailed responses follow.

Reviewer 2 Report

The authors have sufficiently revised the text and some images, improving the work adequately for publication.

Author Response

Thank you for all.

Reviewer 4 Report

I'm satisfied with the revised version of the paper.

Author Response

Thank you for all.